# Investigation into Dynamic Behaviors of High-Temperature Sandstone under Cyclic Impact Loading Using DIC Technology

Hua Lu [1,*], Qiaoli Chen [2,3,*] and Xiaotong Ma [1]

[1] School of Civil Engineering, North Minzu University, Yinchuan 750021, China
[2] College of Civil and Hydraulic Engineering, Ningxia University, Yinchuan 750021, China
[3] Ningxia Center for Research on Earthquake Protection and Disaster Mitigation in Civil Engineering, Yinchuan 750021, China
[*] Correspondence: 2013101@nmu.edu.cn (H.L.); chenqiaoli9110@nxu.edu.cn (Q.C.)

**Abstract:** Coal resources are rich in Ningxia. Long-term mining creates mine goaf, which causes coal to burn spontaneously for a very long time. Unavoidably, the rocks around the coal fire area are affected by high temperatures, which can alter the characteristics of rocks and lead to safety accidents. To explore the temperature influence of sandstone in coal fire areas under cyclic impact loading, the sandstone treated under different temperatures is tested by a split Hopkinson pressure bar (SHPB). The mechanical properties of rocks treated at different temperatures are obtained. The composition of rock is determined, and the energy dissipation is calculated. Meantime, the digital image correlation (DIC) method is applied to study the mechanical behaviors of sandstone. The results show that at the first impact, the peak stress of sandstone decreases as the temperature increases. However, there is no obvious trend in the peak strain. Under the SHPB cyclic impact, the sandstone specimen is completely destroyed after two to three times of impact at different temperatures. At 25~1000 °C, the dynamic peak stress of sandstone decreases with the increase in impact times, and brittle failure occurs. When the impact pressure is 0.6 MPa, the incident energy increases with the impact velocity; the dynamic peak stress increases with the transmitted energy. Using the DIC method, it is found that when the temperature is below 800 °C, the dynamic characteristics of rock specimen have a close correlation with the crack initiation point and extreme point. When the temperature exceeds 800 °C, the rock specimen is seriously damaged, the overall strain is small, and the stress transfer efficiency is low. These findings show that temperature significantly affects the mechanical properties and initial damage of the sandstone, and the performance and damage are abrupt at 800 °C. Meanwhile, the DIC technology can effectively characterize the strain evolution of rock materials and explain the formation and propagation process of cracks, which provides a valid means for studying the damage and crack evolution of materials.

**Keywords:** sandstone in the coal fire area; cyclic impact; energy dissipation; digital image correlation (DIC) technology

## 1. Introduction

With a rapid development of geotechnical engineering, underground engineering, geothermal mining engineering, and large nuclear power plant engineering, blasting excavation is mainly adopted. Due to the requirements of production and technology, it is inevitable to carry out repeated push-type construction. Rock can bear multiple or cyclic dynamic loads. Moreover, the rock in complex environments is not only subject to multiple engineering disturbances, but also its mechanical behavior is often affected by temperature. Rock is a typical multi-phase material, whose mechanical properties are affected by temperature factors.

The Ningxia Rujigou coalfield fire area is located in Pingluo County of Ningxia Hui Autonomous Region and Alxa Left Banner of Inner Mongolia Autonomous Region. Many

fire zones exist in the current mining area because of the underground mining and dis-ordered mining left over by previous history or the connectivity of rock fractures, joints, and coal seams. The combustion depth of the fire zone exceeds 300 m, and the surface temperature is even above 300 °C, which inevitably leads to changes in rock mechanical properties. If the design is carried out according to the original parameters, it is very easy to cause safety accidents under blasting loads. Therefore, research into rock multi-field coupled dynamics is necessary.

Researchers have extensively explored the rock mechanical properties under impact loading [1–10]. For example, Demirdag et al. [11] concentrated on the static and dynamic compression behaviors of rocks by porosity, Schmidt hardness, and unit volume weight. They found that as the porosity decreased, the compressive strength of rocks increased. Li et al. [12] conducted a cyclic impact test of granite at a strain rate of $101/s^{-1}$ using a modified SHPB device. The results show that when the peak stress is 60~70% of the static uniaxial compressive strength, internal damage of rock under cyclic load is less. When the cyclic impact load approaches the static uniaxial compressive strength, two or three cycles of impact may result in a complete failure of specimen. In addition, Jin et al. [13–15] studied the mechanical properties of rock under coupled static and cyclic impact loads, which achieved certain results. Different from the above analysis, Wang et al. [16] conducted XRF and XRD tests on coal mine mudstone to obtain the main components of the specimen and obtained their dynamic mechanical properties, fracture characteristics, and energy dissipation laws. In addition, Mishra et al. [17] studied the dynamic response characteristics of three types of rocks under high strain rates by a SHPB device. According to the static compression test, combined with the X-ray diffraction, the mineral composition and static mechanical parameters of rocks were obtained. Moreover, Yu et al. [18] obtained the dynamic characteristics of sandstone under impact loading by experimental analysis and came to a conclusion that when sandstone is subjected to cyclic loading, its failure is mainly tensile failure. Some researchers analyzed and studied the rock under cyclic impact loading from the aspects of strain rate, dynamic characteristics, and damage performance [19–21].

Temperature is an important factor affecting rock properties. To study the energy dissipation characteristics of heat-treated sandstone under cyclic impact loads, Shu et al. [22] conducted a series of experiments using SHPB. In addition, Hassanzadegan et al. [23] investigated the influence of temperature on the static and dynamic elastic modulus and porosity of sandstone and concluded that high temperature produced new cracks from an analysis of crack porosity. Moreover, Vidana et al. [24] conducted the triaxial tests under different temperatures and different confining pressures and established a constitutive model for the sandstone under high temperature and high pressure. Moreover, Zhang et al. [25] carried out a triaxial compression mechanical experiment of tight sandstone heated from 0 °C to 1000 °C and combined the thermal weight loss, XRD, and SEM to analyze the micro-fabrics characteristics at different temperatures. This further revealed the influence of temperature on the mechanical properties of tight sandstone. Furthermore, Liang et al. [26] studied the mechanical properties of siliceous sandstone after varying temperature treatment and concluded that high temperature remarkably affects the rock composition, strength, and failure characteristics.

In recent years, the digital image correlation (DIC), as a contactless and high-precision optical measurement technique, has been gradually applied in rock mechanics. The DIC technology has made some achievements in rock mechanics for the past few years. Qi et al. [27] investigated the effect of joint density on the strength characteristics and fail-ure mechanism of rock; DIC technology was applied to calculate the strain field of rocks under compression loading. Results showed that the failure mechanism of the specimens was strongly associated with the joint density. Similarly, Munoz et al. [28] used the DIC technology to measure the deformation and strain by performing a static test on sandstone and obtained the strain field on the sandstone surface. Furthermore, Xu [29] used the DIC technology and numerical simulation method to study the nucleation and propagation of a single crack and double cracks of rock-like materials during the uniaxial compression test

and obtained the crack development law. For studying the crack extending and damage deformation characteristics of rock subjected to static loading, some scholars have carried out comprehensive detection and analysis by the DIC technology [30,31]. In the study of rock dynamics, Fourmeau et al. [32] performed a Brazilian disk test of granite by SHPB technique; combined with the DIC technology, they obtained the displacement and strain field of specimen surface. However, Gao et al. [33,34] used the DIC technology to measure cracks in rock notched semi-circular bending test and analyzed the dynamic rock fracture process. They determined the displacement and strain field with crack propagation. Different from the above analysis, Yang et al. [35] conducted a dynamic test of composite rock under dynamic loading at varying impact velocities by SHPB. They also combined with DIC technology to capture the failure process. The test results revealed its dynamic characteristics and failure characteristics.

It can be seen from the above research that most scholars mainly focus on the crack development or damage detection of rock under static or dynamic loading by DIC technology. Unfortunately, few studies can be found on the mechanical properties of rock after high-temperature treatment under impact loading. In coal fire areas, rocks here may be affected by multiple or cyclic dynamic loading. Therefore, it is necessary to discuss the dynamic behavior of rocks in the coal fire areas under cyclic loading.

Under cyclic impact loading, the mechanical characteristics of rocks after high temperatures are studied via a 75-milimeter SHPB equipment in this paper, and the DIC method is adopted to analyze the deformation of sandstone specimens treated at different temperatures during the first impact. The deformation field evolution and crack generation process of sandstone under external forces are obtained, which offer a theoretical support for the safety and post-disaster evaluation of rock engineering.

## 2. Specimen Preparation and Test Equipment

### 2.1. Specimen Preparation

The experiment flow chart is shown in Figure 1. The rock specimens used herein were all taken from the same aperture in the Rujigou mining area of Ningxia, and the drill bit diameter was 70 mm. To ensure the reliability of test results, sandstone with good integrity and homogeneity was used. According to the research on the optimal size of SHPB impact test by scholars, specimens were processed into a 1:1 cylinder, and the two ends of t specimens were polished to control their flatness within 0.02 mm. The average diameter and size of specimens was 67 mm. Figure 2 shows a partially processed sandstone specimen.

After specimens were processed, the selected specimens were classified according to their longitudinal wave velocity. In the cyclic impact test, in order to explore its dynamic characteristics in different temperature environments, sandstone specimens with a similar density and longitudinal wave velocity ($V_P$) were selected. The basic physical parameters of sandstones are listed in Table 1, in which the porosity is determined by the method in the literature [36].

It can be observed from Table 1 that the $V_P$ and density of sandstone specimens are similar. However, the effective porosity is slightly different, which is related to the internal structure of the rock itself. After the porosity of sandstone was tested, the sandstone was dried for more than a week. Then, the specimen was put into a muffle furnace for heat treatment. To reach the predetermined temperature at both inside and outside of a specimen, the specimen was heated to a specified temperature and kept for four hours. Then, it was naturally cooled to room temperature. The heating device is presented in Figure 3, with a size of 600 mm $\times$ 400 mm $\times$ 300 mm, and the maximum temperature that can be heated is 1000 °C.

Figure 4 shows the appearance of sandstone specimens at different temperatures. It can be observed from Figure 4 that with an increase in temperature, the apparent color of sandstone gradually becomes darker from gray at 25 °C. That is, the apparent color of the sandstone changes little at 200 °C and 400 °C, while it gradually becomes dark red at 600 °C and 800 °C, and turns brown at a high temperature of 1000 °C.

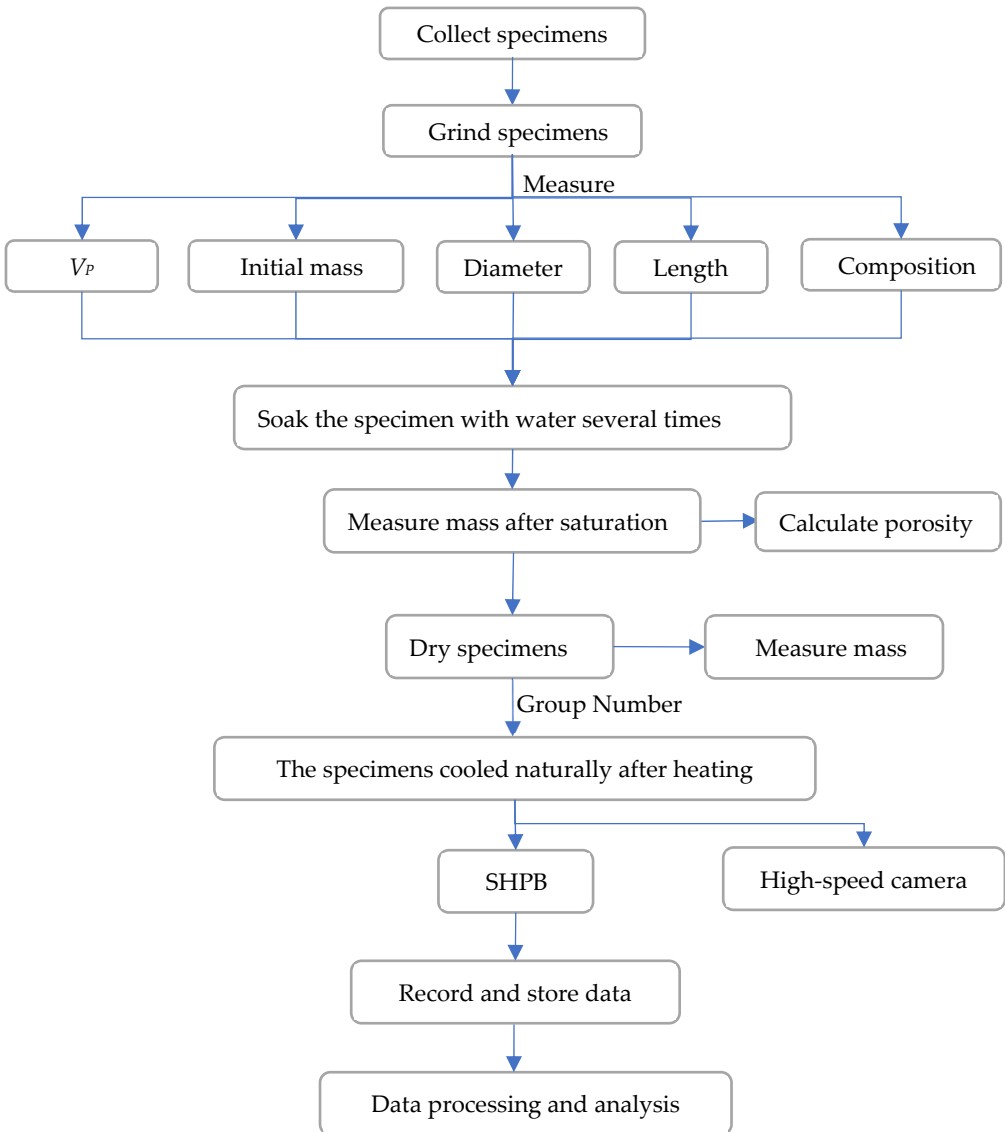

**Figure 1.** Experiment flow chart.

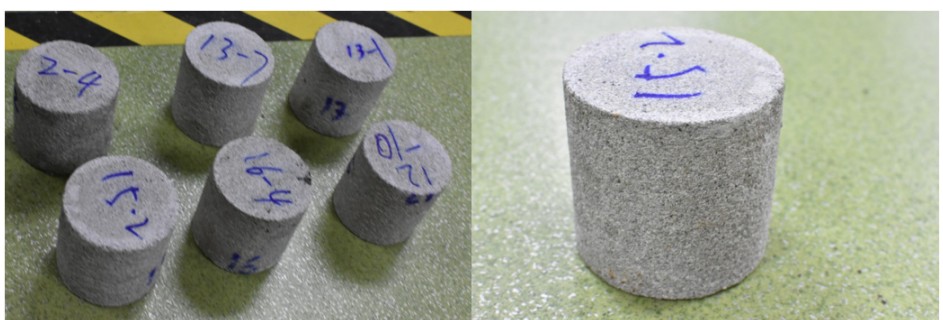

**Figure 2.** Sand specimen after grinding.

**Table 1.** Parameters of specimens.

| Specimen No. | Temperature/°C | Diameter/mm | Length/mm | Initial Mass/g | Porosity/% | Density/kg·m$^{-3}$ | $V_P$/m·s$^{-1}$ |
|---|---|---|---|---|---|---|---|
| 7-1 | 25 | 67.16 | 66.6 | 562.85 | 2.94 | 2385.65 | 2579 |
| 8-5 | 200 | 66.86 | 66.68 | 546.79 | 3.11 | 2335.62 | 2551 |
| 6-3 | 400 | 67.5 | 66.7 | 539.01 | 2.74 | 2258.25 | 2310 |
| 17-2 | 600 | 68.5 | 67 | 562.44 | 3.13 | 2277.87 | 2392 |
| 23-2 | 800 | 67.4 | 67 | 550.24 | 3.21 | 2301.80 | 2554 |
| 4-4 | 1000 | 67.66 | 67.2 | 545.71 | 3.16 | 2258.59 | 2429 |

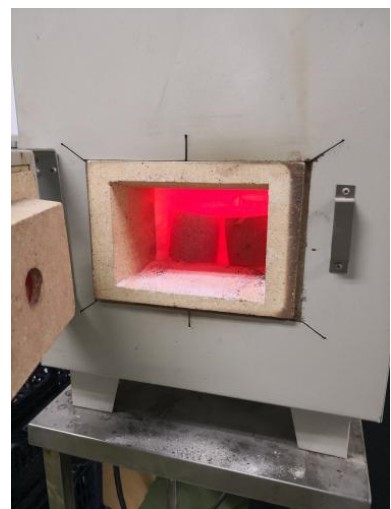

**Figure 3.** Muffle furnace.

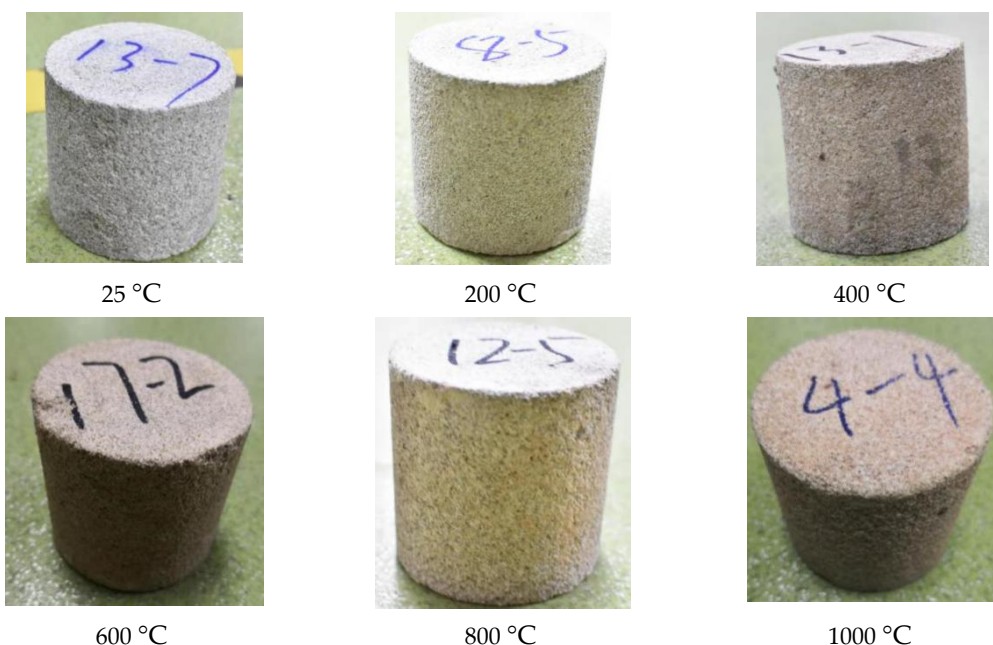

**Figure 4.** The appearance of sandstone specimens at different temperatures.

## 2.2. Impact Test Device

A diameter of 75 mm SHPB test device is given in Figure 5. The length of the cylindrical punch is 400 mm. The diameters of the incident bar, transmitted bar, and absorption bar are all 75 mm. The length, elastic modulus, and density of incident bar and transmitted bar are 2 m, 210 GPa, and 7800 kg/m$^3$, respectively.

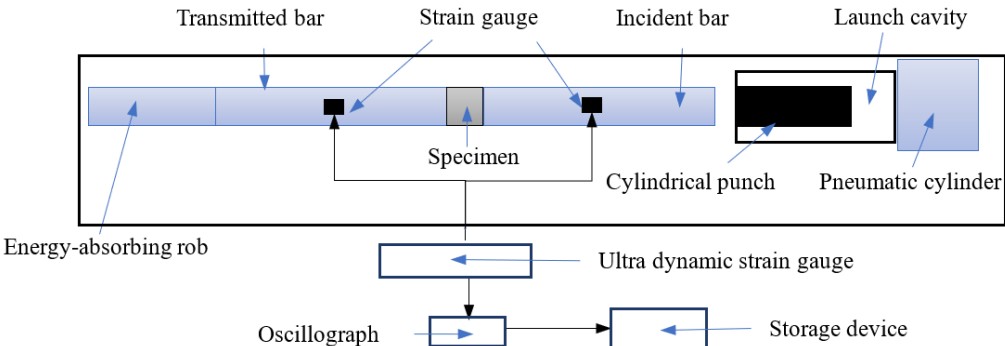

**Figure 5.** SHPB test system.

Figure 6 shows the placement of sandstone specimens during impact test. According to one-dimensional and stress uniformity assumption, when the bullet in the launch cavity hits the incident bar, stress wave propagates in the incident bar and reaches the contact surface between the incident bar and specimen. At this time, a part of the stress wave is reflected, and the other part of the stress wave passes through the specimen and reaches the contact surface between the specimen and transmitted bar. Wave refracts back and forth in the specimen until the stress on both ends of the specimen tends to be uniform.

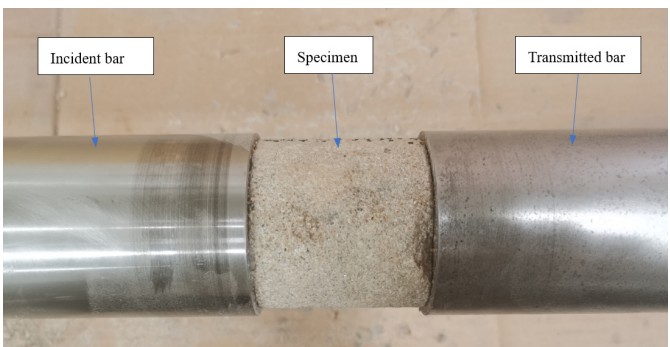

**Figure 6.** Placement of sandstone specimens.

In the SHPB impact test, the impact pressure was set to 0.6 MPa. Under the equal-amplitude cyclic impact loading, sandstone specimens with six temperatures of 25 °C, 200 °C, 400 °C, 600 °C, 800 °C, and 1000 °C were tested. Figure 7 shows a typical dynamic stress equilibrium curve, in which the superposition of In (incident strain) and Re (reflected strain) coincides with the Tr (transmitted strain). It can be observed that the two ends of sandstone meet the stress balance conditions and the basic assumptions.

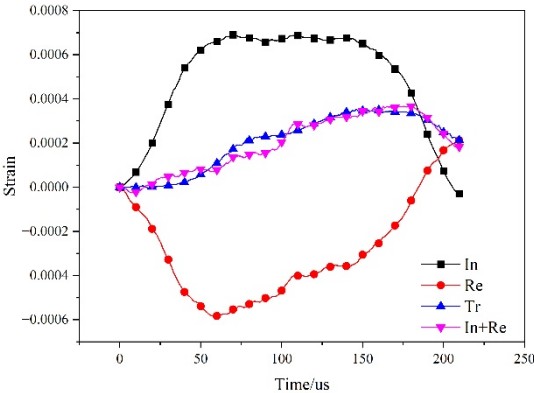

**Figure 7.** Verification of dynamic force balance of the sandstone.

## 3. Analysis of Dynamic Performance and Energy Consumption

### 3.1. Mineral Composition Analysis at Different Temperatures

The mineral composition of sandstone specimens treated at 25 °C and different high temperatures was analyzed by the XRD technology. The results showed that the main components of sandstone in coal fire area were quartz ($SiO_2$), potassium feldspar ($KAlSi_3O_8$), and mica ($KAl(AlSi_3)O_{10}(OH)_2$), as shown in Figure 8. In the XRD technology, the peak position is determined by the size and shape of a unit cell, and the peak intensity is determined by the type and original position of atoms in the unit cell. Therefore, for the mixture, the peak diffraction intensity mainly increases with the substance content. The peak diffraction intensity of quartz after treatment at different temperatures is the highest in Figure 8, which indicates that the content of quartz is the most. Compared with the sandstone specimens at 25 °C, the peak diffraction intensity of quartz decreases slightly with an increase in temperature. Some studies [37] have shown that quartz has a strong ability to absorb the X-rays, and its peak diffraction intensity is relatively high, which can easily cover up other mineral components. Meantime, the fusion and boiling point of quartz are as high as 1723 °C and 2230 °C, respectively. Therefore, the influence of high temperature on quartz is not obvious. Under high temperature, other clay minerals in the sandstone lose free water to interlayer water and structural water, and their swell ability under high temperatures has a significant effect on the mechanical characteristics of the sandstone.

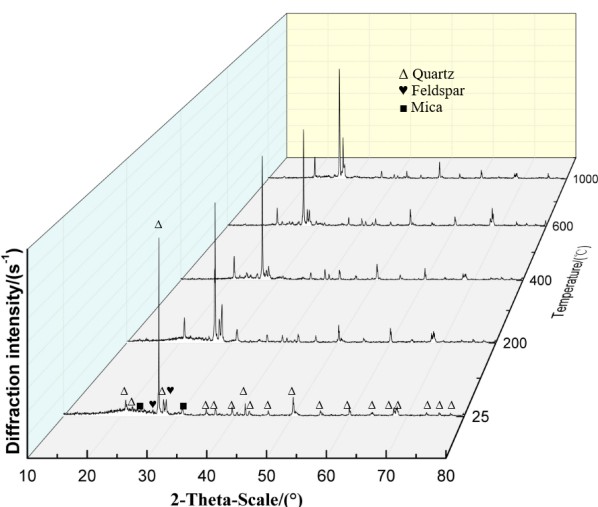

**Figure 8.** XRD spectra of sandstone at different temperatures.

### 3.2. Analysis of Dynamic Mechanical Properties in the First Impact

During the test, the impact velocity is not only affected by the impact pressure, but also by the position of the bullet. Hence, under the same impact load, a certain difference occurs in the impact velocity. The cyclic impact test conditions and results of sandstone after different high temperature treatment are listed in Table 2.

Figure 9 gives the relationship between the dynamic peak stress, peak strain, and temperature of the sandstone specimen at the first impact. It can be concluded from Table 2 and Figure 9a that when the temperature increased, the dynamic peak stress of sandstone specimen at the first impact gradually decreased. Furthermore, the dynamic peak stress under the first impact is the largest at 25 °C. In addition, the strength loss at 200 °C and 800 °C is greater than those at other temperatures, and the decline rate reaches 37.7% and 49.9% compared with the strength at 25 °C and 600 °C, respectively. Meanwhile, as presented in Figure 9b, the peak strain decreased significantly at 200 °C and 800 °C, corresponding to the change in peak strength with time. At 25 °C and 200 °C, the ductility of the sandstone is small, and the sandstone is brittle. Therefore, the peak strength and peak strain greatly decreased. Figure 9b,c show that after experiencing high temperatures

of 400 °C and 600 °C, the sandstone loses interlayer water and part of its structural water. The pores increase, so the peak strength decreases, while the peak strain and average strain rate increase. The sandstone loses the structural water under high temperatures of 800 °C and 1000 °C, the cementitious material changes, and the micro-cracks in the sandstone increase, resulting in a serious embrittlement. As a result, the peak strength, peak strain, and average strain rate decrease sharply. The above results show that high temperature causes a certain degree of heat loss to the sandstone specimen. That is, with an increase in temperature, the water in the sandstone evaporates, the mineral composition changes, the cohesive force decreases, and the specimen is easily broken.

Figure 10a gives the stress–strain curve of sandstone at 25 °C at the first impact. It can be observed that the dynamic stress–strain curve can be divided into the following stages. The OA section is the compaction section, with a short action time and a rapid increase in stress, and the initial dynamic modulus are the largest. The AB section, which is basically linear, can be approximately regarded as the elastic stage, and the elastic modulus decreases compared with that of the OA section. Stress in the BC section increases rapidly, which is an upward convex curve. In this section, a strain hardening occurs. The stress growth in the CD section begins to slow down, but the strain increases rapidly. That is, a strain softening occurs. The pre-peak stage of DE reflects certain plastic characteristics of rock. The elastic modulus decreases, but strength hardens. Point E is the peak strength of the first impact. The BE section reflects the fracture development stage of the sandstone. The EF section is the post-peak stage, and its decline rate reflects the plasticity of the sandstone. Figure 10b gives the stress–strain curves of sandstone at varying temperatures. The first impact stress–strain curves after different high-temperature treatment show different characteristics. The ductility and brittleness of sandstone at 200 °C increase due to water loss as the temperature increases. The boundary between the compaction section and the elastic section is not obvious, the initial dynamic modulus is the largest, and the peak stress and peak strain decrease significantly. The sandstone recrystallizes at 400 °C and 600 °C, the micro-cracks increase, the sandstone exhibits obvious plastic characteristics, and the peak strain increases. In addition, under high temperatures of 800 °C and 1000 °C, the sandstone is seriously embrittled, the mineral cohesion is reduced, and the strength and strain greatly decrease.

**Table 2.** Cyclic impact test conditions and results of sandstone after different temperatures.

| Specimen No. | Temperature/°C | Impact No. | Impact Velocity/m·s⁻¹ | Peak Stress/MPa | Peak Strain | Average Strain Rate/s⁻¹ |
|---|---|---|---|---|---|---|
| 7-1 | 25 | 1 | 7.81 | 56.86 | 0.00635 | 45.64 |
| | | 2 | 8.23 | 57.53 | 0.00486 | 50.97 |
| | | 3 | 8.05 | 41.63 | 0.00545 | 47.57 |
| 8-5 | 200 | 1 | 7.90 | 35.45 | 0.00335 | 70.95 |
| | | 2 | 7.84 | 38.98 | 0.00497 | 60.93 |
| | | 3 | 7.58 | 37.21 | 0.00684 | 58.19 |
| 6-3 | 400 | 1 | 7.78 | 34.24 | 0.00648 | 51.23 |
| | | 2 | 7.99 | 29.29 | 0.00178 | 53.29 |
| | | 3 | 7.87 | 20.17 | 0.00018 | 57.37 |
| 17-2 | 600 | 1 | 8.23 | 29.76 | 0.00733 | 67.31 |
| | | 2 | 7.90 | 27.90 | 0.00129 | 67.42 |
| 23-2 | 800 | 1 | 8.47 | 14.92 | 0.00055 | 30.95 |
| | | 2 | 8.08 | 10.89 | 0.00291 | 59.73 |
| 4-4 | 1000 | 1 | 7.84 | 8.66 | 0.00080 | 38.24 |
| | | 2 | 8.05 | 6.56 | 0.00024 | 22.74 |

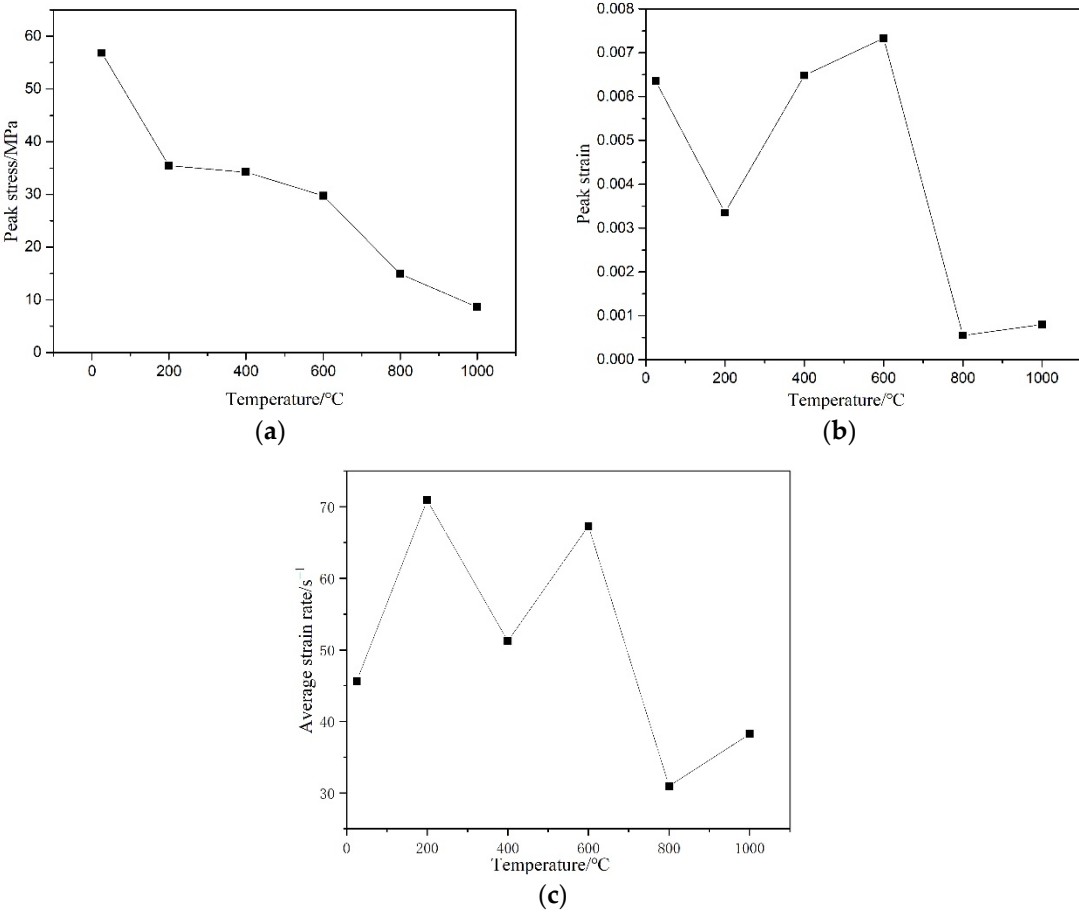

**Figure 9.** Relationship between first impact dynamic peak stress, peak strain, average strain rate, and temperature of sandstone. (**a**) Relationship between peak stress and temperature. (**b**) Relationship between peak strain and temperature. (**c**) Relationship between average strain rate and temperature.

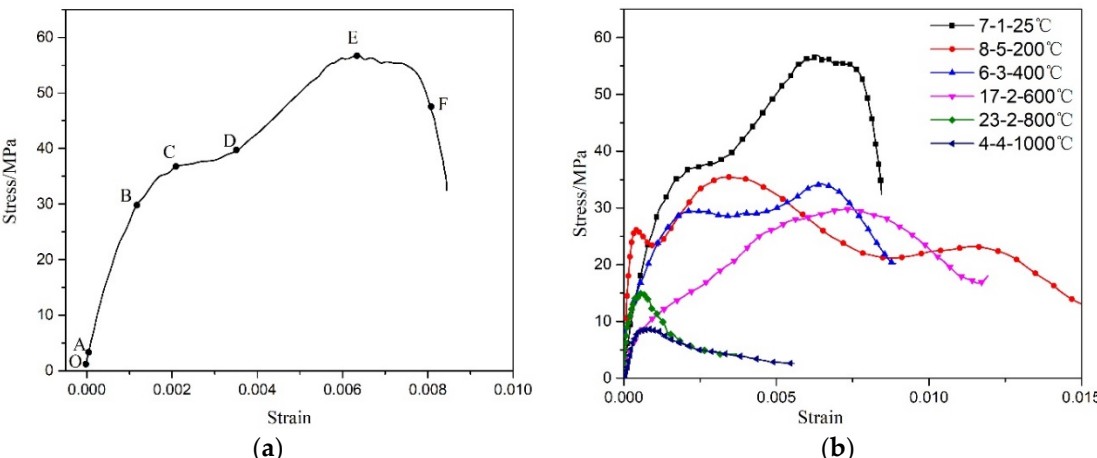

**Figure 10.** Stress–strain curves of sandstone in the first impact. (**a**) At room temperature. (**b**) At different temperatures.

### 3.3. Analysis of Cyclic Impact STRESS and Strain

In this paper, under the same impact load, sandstone specimen is subjected to the equal-amplitude cyclic impact until a complete destruction. In the test, the original data obtained by the ultra-dynamic strain gauge were processed, and the stress and strain were calculated using a 'two-wave method' formula [38]. The stress–strain curves of

the sandstone under cyclic impact at different temperatures were obtained, as shown in Figure 11.

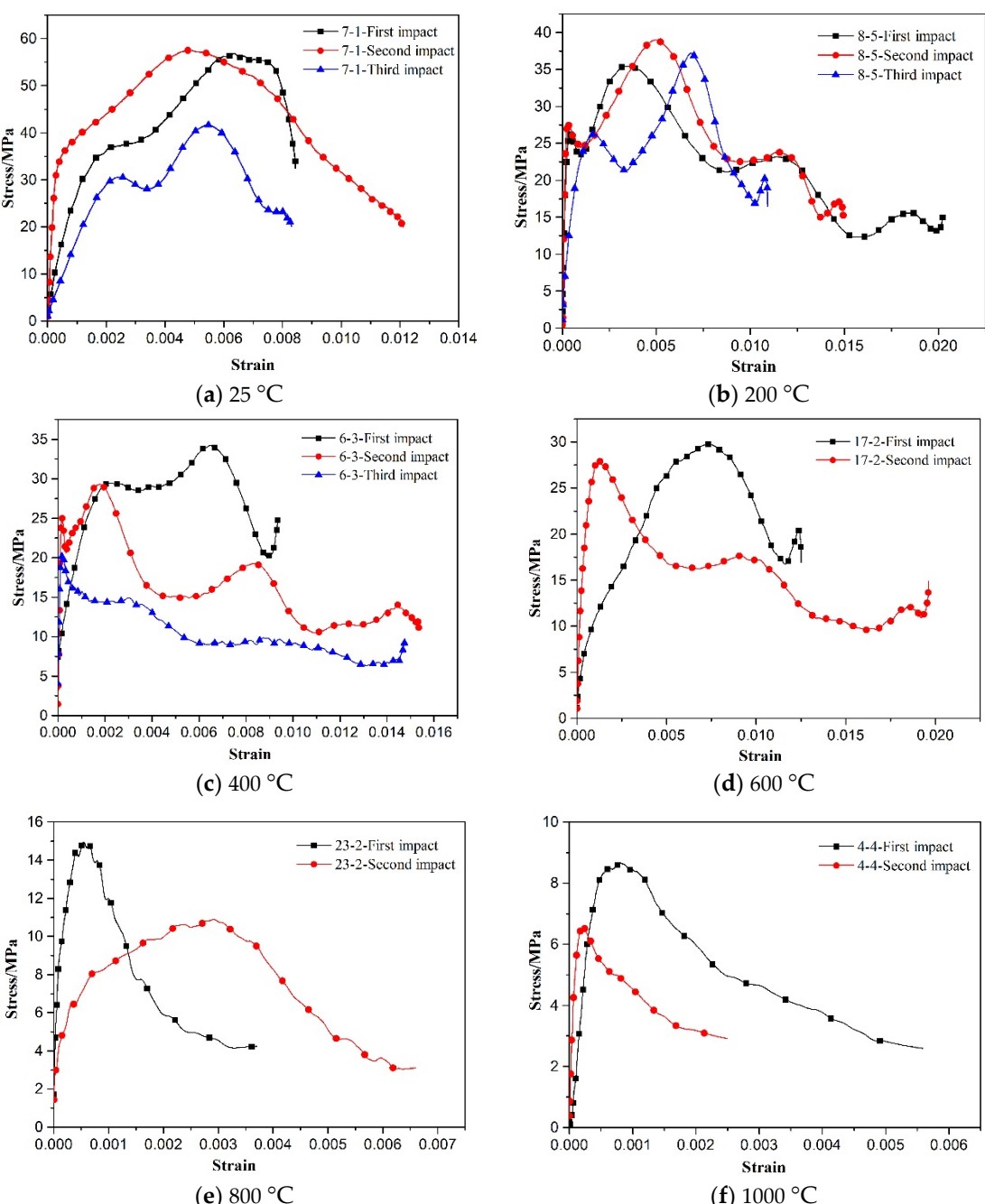

**Figure 11.** Cyclic impact stress–strain curves of sandstone after different temperatures.

The cyclic impact stress–strain curves of sandstone specimens after different high-temperature treatments are plotted in Figure 11. From Table 2 and Figure 11, it is observed that when the impact pressure is set to 0.6 MPa, the sandstone specimens are completely destroyed after two to three cycles of impact. It can be concluded from Figure 11 that, at different temperatures, the stress–strain curves of sandstone specimens after the second impact show obvious brittle characteristics compared with those after the first impact. In the second impact, the compaction and elastic stages are significantly shortened, and the initial elastic modulus significantly increases. It shows that the internal pores are compressed under the first impact, and the initial deformation is small under impact loading,

but the stress increases rapidly. At 800 °C, the sandstone curve presents different variation laws. This is perhaps because large cracks appear in the sandstone under the first impact and the sandstone specimen is already in a failure state. Therefore, under the second impact, cracks in the sandstone develop rapidly and the deformation increases, showing an obvious plastic property. In addition, at 25 °C, the sandstone specimen is damaged three times by cyclic impact. Due to a development of cracks, the compaction and elastic stages of sandstone under the third impact are significantly larger than those of the second impact, which indicates that the sandstone has experienced the stages of pore compaction, crack development, and failure under cyclic loading. The results show obvious plastic characteristics. In addition, for the sandstone specimen after 200 °C, as analyzed earlier, the sandstone loses water, the ductility increases, and the high temperature causes a certain heat loss, resulting in a decrease in the strength. The thermal stress produces fewer cracks in the sandstone. Therefore, the dynamic characteristics under cyclic loading are similar to those under 25 °C, which shows a certain plasticity. Moreover, for the sandstone specimens with high temperatures of 400 °C and 600 °C, with an increase in heat treatment temperature, the brittleness of sandstone increases under repeated loads, indicating that the number and length of micro-cracks increase with the temperature. Therefore, the cracks generated by high temperature at the first impact are developed. In the subsequent impact, when temperature increases, the plasticity of sandstone decreases and the brittleness increases, resulting in an impact failure of the sandstone. When the temperature increases from 800 °C to 1000 °C, the thermal expansion of particles, thermal cracking, and clay mineral morphology of the sandstone change, leading to a cohesive failure and a development of micro-cracks in the sandstone. Thus, the strength decreases sharply and brittle failure occurs.

### 3.4. Analysis of Cyclic Impact Energy

After the sandstone specimen is subjected to cyclic dynamic impact, its internal damage gradually increases. The initiation, development, propagation, and penetration of cracks all need to absorb energy from the outside. The laws of thermodynamics show that energy transformation is the intrinsic nature of a change in physical characteristics and that the internal damage of rock is closely related to energy dissipation. Based on the energy conservation law and one-dimensional stress wave theory, the incident energy, reflected energy, and transmitted energy can be obtained by Equation (1) to Equation (3).

$$W_I = \frac{AC}{E} \int_0^t \sigma_I^2(t) dt \tag{1}$$

$$W_R = \frac{AC}{E} \int_0^t \sigma_R^2(t) dt \tag{2}$$

$$W_T = \frac{AC}{E} \int_0^t \sigma_T^2(t) dt \tag{3}$$

where $W_I$, $W_R$, and $W_T$ represent the incident energy, reflected energy, and transmitted energy, respectively; A, C, and E are the cross-sectional area, longitudinal wave velocity, and elastic modulus of the pressure bar, respectively; $\sigma_I^2(t)$, $\sigma_R^2(t)$, and $\sigma_T^2(t)$ are incident stress, reflected stress, and transmitted stress at time $t$, respectively.

$$W_L = W_I - W_R - W_T \tag{4}$$

$$\zeta = W_L / W_I \tag{5}$$

The dissipated energy and dissipated energy ratio of sandstone specimens can be obtained according to Equation (4) and Equation (5). The calculations of the cyclic impact energy of sandstone after different high-temperature treatments are presented in Table 3 and Figure 12. Four conclusions can be achieved.

**Table 3.** Calculation of cyclic impact energy of sandstone after different temperatures.

| Specimen No. | T/°C | Impact No. | Peak Stress/MPa | Impact Velocity/m·s$^{-1}$ | $W_I$/J | $W_R$/J | $W_T$/J | $W_L$/J | $\zeta$/% |
|---|---|---|---|---|---|---|---|---|---|
| 7-1 | 25 | 1 | 56.86 | 7.81 | 307 | 132 | 36 | 139 | 45.3 |
| | | 2 | 57.53 | 8.23 | 357 | 246 | 40 | 71 | 19.9 |
| | | 3 | 41.63 | 8.05 | 342 | 136 | 26 | 180 | 52.6 |
| 8-5 | 200 | 1 | 35.45 | 7.90 | 318 | 215 | 16 | 87 | 27.4 |
| | | 2 | 38.98 | 7.84 | 318 | 257 | 19 | 42 | 13.2 |
| | | 3 | 37.21 | 7.58 | 322 | 149 | 16 | 157 | 48.7 |
| 6-3 | 400 | 1 | 34.24 | 7.78 | 332 | 164 | 23 | 145 | 43.7 |
| | | 2 | 29.29 | 7.99 | 333 | 248 | 12 | 73 | 21.9 |
| | | 3 | 20.17 | 7.87 | 313 | 235 | 5 | 73 | 23.3 |
| 17-2 | 600 | 1 | 29.76 | 8.23 | 375 | 217 | 15 | 143 | 38.2 |
| | | 2 | 27.90 | 7.90 | 353 | 287 | 8 | 58 | 16.4 |
| 23-2 | 800 | 1 | 14.92 | 8.47 | 489 | 423 | 16 | 50 | 10.2 |
| | | 2 | 10.89 | 8.08 | 351 | 218 | 0.4 | 132.6 | 37.8 |
| 4-4 | 1000 | 1 | 8.66 | 7.84 | 346 | 313 | 0.6 | 32.4 | 9.4 |
| | | 2 | 6.56 | 8.05 | 356 | 320 | 0.3 | 35.7 | 10.02 |

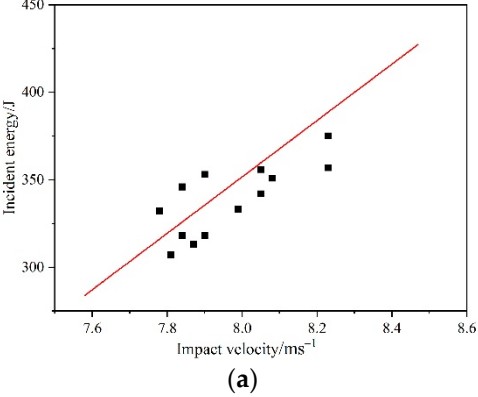

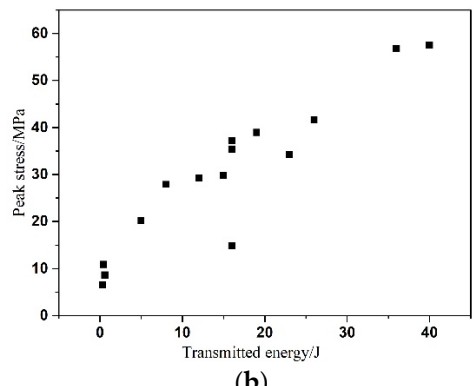

(**a**)  (**b**)

**Figure 12.** Analysis of cyclic impact energy of sandstone. (**a**) Relationship between impact velocity and incident energy. (**b**) Relationship between peak stress and transmitted energy.

(1) When the impact velocity increases, the incident energy increases gradually and is approximately linearly distributed. The peak stress increases with the transmitted energy.

(2) When T ≤ 600 °C, the energy dissipated ratio of the first impact is greater than that of the second impact. According to the energy conservation law, the kinetic energy of sandstone specimen is converted into the energy consumption and other loss energy absorbed by sandstone fracture during impact. For the dissipated energy, due to a deformation of pore body, micro-plastic deformation of the solid medium occurs around the pore and there is a rupture of the solid medium. When the stress wave around the pore passes through the pore body, part of the energy is consumed. This indirectly reflects the damage failure mechanism inside the solid and further indicates that the rock deformation failure is a comprehensive result of energy release and energy dissipation.

(3) When the temperature is 25 °C, 200 °C, and 800 °C, the dissipated energy of the sandstone under the last impact is far greater than that under the previous impact. The reason may be that there are many cracks in the sandstone in the previous impact, but they are not completely destroyed. At the last impact, a large amount of energy is lost from cracks, and the reflected energy and transmitted energy actually acting on the sandstone are small, resulting in an increase in dissipated energy.

(4) At the first impact, the dissipation energy of sandstone decreases sharply at treatment temperatures of 800 °C and 1000 °C, indicating that high-temperature thermal damage had

caused cracks in the sandstone, and the internal damage is serious. The sandstone was destroyed by impact loads, and the absorbed energy was reduced. That is, the dissipated energy was reduced.

## 4. Analysis of DIC Results at the First Impact

Compared with that in the static test, the impact loading velocity in the SHPB dynamic test is large, and the specimen failure duration is short. Therefore, a high-speed camera with a resolution of $512 \times 256$ and a shooting rate of 50,000 fps is applied to capture the whole loading process of sandstone. A high-power LED photography lamp is used to illuminate the signal surface of specimens. To obtain the sandstone deformation during the whole loading, special personnel shall operate the loading equipment and LED photography lamp to ensure that the loading, photography, and lighting in a test are strictly synchronized. The DIC is a technique that tracks the strain and displacement changes on the sandstone surface. Therefore, random dots on the surface of the specimen are required as the speckle patterns before an experiment, as presented in Figure 13. During the loading process, the speckle moves with the deformation of the specimen. Then, the strain field in the specimen is given by analyzing the motion of the speckle.

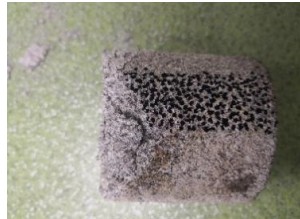

**Figure 13.** Surface speckle pattern of specimen.

Figure 14 gives the vertical strain clouds of sandstone after varying temperature treatments under impact loading. Figure 15 gives the maximum tensile strain at the corresponding loading period of sandstone specimens after varying temperature treatments in Figure 14. The following five conclusions can be obtained.

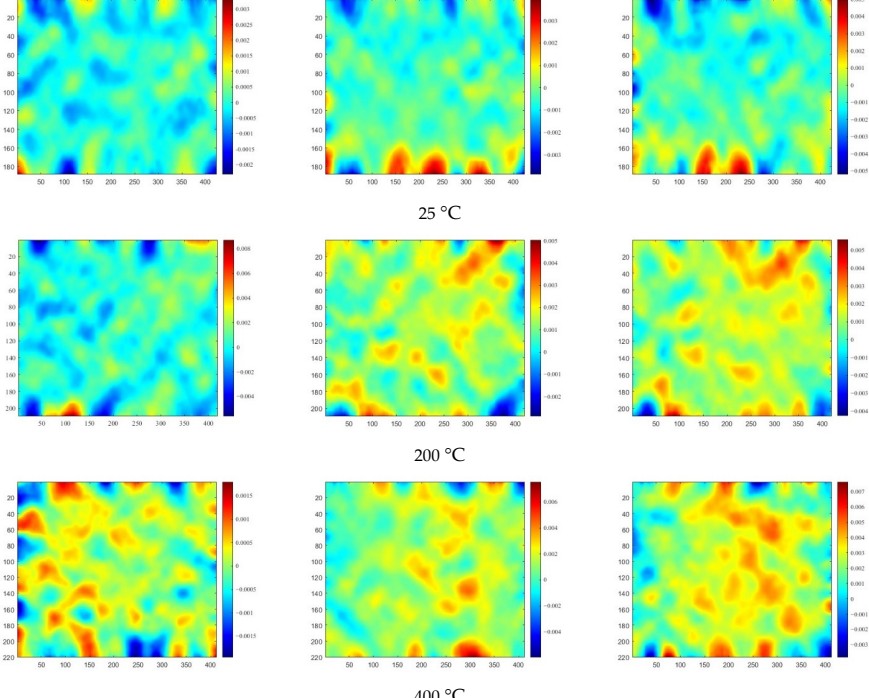

**Figure 14.** *Cont.*

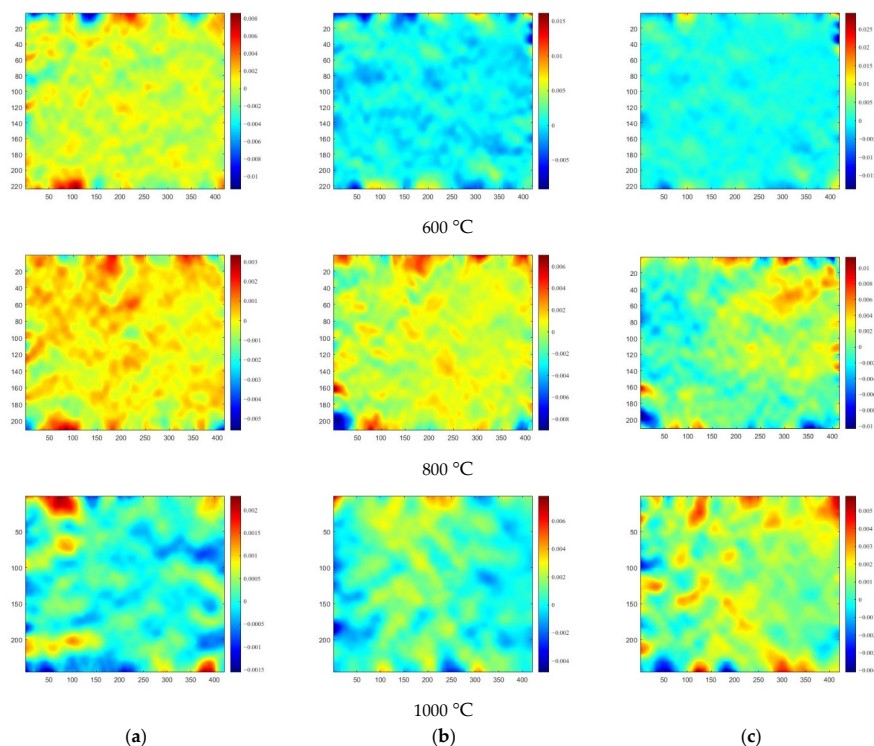

**Figure 14.** Vertical strain clouds of sandstone specimens after different temperatures under impact loading. (**a**) Initial loading period. (**b**) Mid-loading period. (**c**) Later loading period.

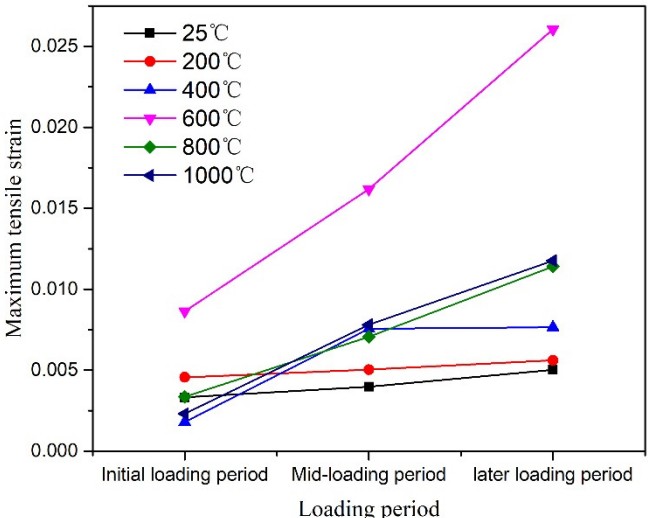

**Figure 15.** Maximum tensile strain during impact loading.

(1) In the initial stage of loading, when the strain field in the vertical direction is at 25 °C and 200 °C, the distribution of the strain field on specimen surface is relatively uniform, and the average tensile and compressive strains of the specimen are small. In addition, the tensile strain distribution area of sandstone specimens from 400 °C to 800 °C gradually increases, and the tensile strain value of the sandstone at 600 °C is significantly larger than that of the sandstone at other temperature sections.

(2) In the middle stage of loading, with a continuous increase in dynamic loading force in the vertical direction of sandstone at 25 °C, a local hardening zone appears on the edge of specimens. However, the value is small and no significant change occurs in the strain until the later loading stage. It shows that the overall strength of sandstone at 25 °C is high under impact loading, and the sandstone does not have overall cracks.

(3) When temperature T = 200 °C, the tensile strain distribution area of the sandstone increases significantly in the middle of loading. At the initial loading stage, an area of concentrated tensile strain appears. Then, the strain gradually expands until the later loading stage. The lateral strain region parallel to the loading direction evolves into an inclined shear strain region and the tensile strain in this region changes little. The literature [35] has defined two strength values: the initial strength value and the maximum strength value, which correspond to the strength value when the rock begins to fail and the maximum value of the stress time history curve, respectively. It is pointed out that the point corresponding to the crack initiation time is an inflection point of the stress time history curve, but it is not the extreme point. Therefore, compared with that in Figure 10b, the strength of sandstone at 200 °C changes little from the initiation point to the extreme point. The strain development of sandstone at 400 °C is almost the same to that of the sandstone at 200 °C. In short, the DIC technology analysis can explain the dynamic characteristics and failure mode of sandstone under high temperatures.

(4) When T = 600 °C, due to a significant increase in micro-cracks under high temperature, almost the entire area is tensile strain in the vertical direction at the initial loading stage. Its tensile strain value is much bigger than that of the sandstone at other temperature sections in the middle and later loading stages. This finding reflects that the sandstone is subjected to an external force and the internal micro-cracks evolve continuously from disordered distribution to orderly development, showing obvious plastic features.

(5) The failure characteristics of the sandstone at 800 °C and 1000 °C are similar, both of which are reflected in the presence of micro-cracks inside the sandstone after high-temperature treatment. Different from that at 600 °C under impact loading, the stress of sandstone changes little. It indicates when the high temperature is 800 °C and 1000 °C, the initial internal damage of sandstone is serious and the brittleness is large. Under impact loading, the stress is less than downward transmission so the strength and strain are very small.

## 5. Conclusions

The sandstone specimens subjected to different temperatures were cyclically impacted by an SHPB test equipment. The physical and mechanical behaviors of sandstone were analyzed from the perspectives of mineral composition, stress–strain, and energy, and the conclusions are as follows.

(1) With an increase in temperature, the peak stress of sandstone shows a downward trend in the first impact. The peak strength of sandstone specimens at 200 °C and 600 °C decreases significantly, and the peak strain of sandstone specimens at 200 °C decreases significantly. The sandstone treated at 400 °C and 600 °C show plastic characteristics significantly and the peak strain increases. However, the sandstone specimens at 800 °C and 1000 °C are seriously brittle and the peak strain decreases significantly.

(2) Under the cyclic impact loading, the sandstone specimens were completely destroyed after two to three times of impact. Most sandstone specimens showed obvious brittleness in the second impact. The compaction and elastic stages were obviously shortened and the initial elastic modulus significantly increased. Compared with the early impact, the third impact compaction and elastic stages increased significantly, showing the obvious plastic characteristics.

(3) From energy analysis, when the impact velocity increases, the incident energy also increases gradually and is approximately linearly distributed. The peak stress increases with the transmitted energy. When T ≤ 600 °C, the energy dissipated ratio of the first impact is greater than that of the second impact. At the first impact, the dissipation energy of sandstone decreased sharply at 800 °C and 1000 °C. Changes in energy also reflect the damage in sandstones.

(4) From analysis of DIC results in the first impact in the initial stage of loading, when the strain field in the vertical direction is at 25 °C and 200 °C, the average tensile and compressive strains of the specimen are small. When T = 200 °C and 400 °C, the

tensile strain distribution area of the sandstone increases significantly in the middle of loading. When T = 600 °C, the tensile strain value of the sandstone is greater than that of the sandstone at other temperature sections. When T = 800 °C and 1000 °C, the initial internal damage of sandstone is serious and the stress of sandstone changes little. Therefore, the strength and strain are very small.

**Author Contributions:** Conceptualization, Q.C. and H.L.; methodology, H.L.; software, X.M. and Q.C.; vali-dation, X.M.; formal analysis, X.M.; investigation, X.M.; resources, H.L.; data curation, Q.C.; writing—original draft preparation, Q.C.; writing—review and editing, H.L. and Q.C.; visualization, Q.C.; supervision, H.L.; project administration, H.L.; funding acqui-sition, H.L. All authors have read and agreed to the published version of the manuscript.

**Funding:** This work was financially supported by the Ningxia Natural Science Foundation of China (No.2020AAC03045); Ningxia Key Research and Development Program of China (Grant No. 2021BEG03022); Outstanding Young Teachers Training Foundation of Ningxia of China (NGY2020054); West Light Foundation of The Chinese Academy of Sciences (XAB2021YW14); Ningxia Outstanding Talent Support Program Project of China (Grant No. TJGC2019001), and these supports are gratefully acknowledged.

**Institutional Review Board Statement:** Not applicable.

**Informed Consent Statement:** Not applicable.

**Data Availability Statement:** The data used to support the findings of this study are available from the corresponding author upon request.

**Conflicts of Interest:** The authors declare no conflict of interest.

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
