# Peer review of "Investigation into Dynamic Behaviors of High-Temperature Sandstone under Cyclic Impact Loading Using DIC Technology"

_applsci, doi:10.3390/app12189247_

Round 1

Reviewer 1 Report

1.      In the abstract the problem statement and the objective of the work is not well spell-out. Kindly, restructure the abstract and make it to be in present tense.

2.      Improve the magnification of the figure 2 and give little description of the Muffle furnace

3.      Include the flow chart of the experiment for better clarity and to enhance the manuscript

4.      In section 3.1 for proper identification of the minerals the authors need to include XRF to show the chemical composition and XPS analysis this will show elemental and chemical composition of the sample used.

5.      The Turnitin is 20% kindly reduced it below 15%

6.      Kindly check and correct little grammatical mistakes

7.      Very long sentences are used in the manuscript, which confuse & obscure.

8.      The authors should use a maximum of 11 words per sentence. In this way, the authors can be clear and concise.

Reviewer 2 Report

This manuscript is very interesting for national and international readers. I recommend for publication after minor English Language corrections.

Reviewer 3 Report

The manuscript focuses on the mechanical response of a sandstone subjected to dynamic loading and different temperature. The rock response is analyzed with XRD and DIC technologies. The subject of the manuscript is interesting.

The Introduction must be improved. The subject of the manuscript is reported in a wide international literature. However, the citations belong to the Chinese scientific literature (with several papers/Msc-PhD thesis  written in Chinese) in the percentage 60%. These papers are unknown in the international context, because the universities of the other countries do not have the subscriptions to these databases. In the end, the background cannot be evaluated and the references are not adequate. I think that the authors should make an effort to place their study in an international context.

The term rock engineering is used improperly in two sentences of the Introduction: “Rock engineering can bear multiple or cyclic dynamic loads” and “Besides, Jin et al. [6-8] researched the mechanical properties of rock engineering under coupled static and cyclic impact loading, which achieves certain results”. I think that the term should be simply substituted by “rock”.

The DIC technology is described in the first 16 lines in general terms and in a non-effective way.

The sentence “Some researchers analyzed and studied the rock under cyclic impact loading from the aspects of strain rate, dynamic characteristics, and damage performance [13-15]” is debatable, because there is a wide international literature on these aspects. The authors cited 2 papers of the Chinese literature and 1 PhD thesis written in Chinese.

The sentence  Table 1 lists the basic physical parameters of the sandstones used in the test, in which the effective porosity is determined according to the method in literature [31]”  is positioned too early, because the effect of the temperature has not yet been described. Consequently the steps of the tests are not clear. Furthermore, the sentence on the porosity is debatable, because the reference is not ISRM or equivalent International society.

The sentence “The internal damage of rock varies from generation to expansion with a difference in cyclic impact times” is not clear (from generation to expansion ?).

The sentence “Therefore, the waveform amplitude has a certain difference” is not clear. Maybe: “Therefore, the waveform amplitude induces  a certain difference in the rock response”?

The results of the experiments should be compared with similar experiments found in the international literature. For instance, the stress strain behavior at different temperatures can be compared with the results of static tests, by evidencing the differences that can be attributed to the strain rate effect.

The effect of temperature on the mechanical response of sandstones is well investigated in the international literature. I think that some comments related to the effect of temperature on sandstones are necessary.

The effect of the strain rate is not addressed. I think that a comment should be reported, given the wide international literature on this subject.

Round 2

Reviewer 1 Report

The authors have address all the necessary comments at this stage the paper is read for publication